# Research Progress on the Effect of *Thesium chinense Turcz.* on Neurodegenerative Diseases

**DOI:** 10.3390/ijms26157079

**Published:** 2025-07-23

**Authors:** Ziyi Li, Yanfang Zhao, Rong Wang, Ruoxuan Zhou, Xuehua Chen, Jingchen Jiang, Yilan Dai, Huaiqing Luo

**Affiliations:** 1Department of Physiology, School of Basic Medical Sciences, Hunan Normal University, Changsha 410013, China; ziyi.li@hunnu.edu.cn (Z.L.); zhaoyanfang@hunnu.edu.cn (Y.Z.); zhouruoxuan@hunnu.edu.cn (R.Z.); chenxuehua@hunnu.edu.cn (X.C.); jiangjingchen@hunnu.edu.cn (J.J.); 2Department of Pathophysiology, School of Basic Medical Sciences, Hunan Normal University, Changsha 410013, China; wangrongrongzyk@163.com; 3College of Life Sciences, Hunan Normal University, Changsha 410013, China; yilandai@hunnu.edu.cn

**Keywords:** *Thesium chinense Turcz.*, neurodegenerative diseases, antioxidant, anti-inflammatory, multi-target therapy

## Abstract

*Thesium chinense Turcz.*, a traditional Chinese medicinal herb, is enriched with bioactive constituents such as flavonoids and polysaccharides, demonstrating multifaceted therapeutic properties including anti-inflammatory, antioxidant, and neuroprotective effects. This review systematically elucidates the regulatory mechanisms by which active components of *Thesium chinense Turcz.* modulate pathological processes in NDDs, such as neuroinflammation and oxidative stress. Furthermore, it synthesizes evidence of its neuroprotective efficacy across experimental models and evaluates its translational potential for clinical applications. By integrating preclinical findings and mechanistic insights, this work provides a robust theoretical foundation for advancing natural product-based therapeutics in the management of NDDs.

## 1. Introduction

Neurodegenerative diseases (NDDs), characterized by progressive neuronal degeneration and demyelination, represent a growing global health challenge marked by deteriorating cognitive and physical functions [1]. Prominent NDDs such as Alzheimer’s disease (AD), Parkinson’s disease (PD), Huntington’s disease (HD), and amyotrophic lateral sclerosis (ALS) collectively impact millions worldwide [2]. In China, the aging population has amplified NDD prevalence, with epidemiological data revealing 16,990,827 documented cases of AD and related dementias in 2021, as reported in the China Alzheimer’s Disease Report 2024 [3]. Concurrently, the Chinese Parkinson’s Disease Treatment Guidelines (Fourth Edition, 2020) indicate a 1.7% PD prevalence among citizens aged ≥ 65 years, projecting a national caseload of 5 million by 2030—nearly half of the global burden. These disorders inflict multidimensional harm: progressive cognitive decline, motor impairment, and behavioral disturbances erode patients’ independence and quality of life, while escalating healthcare costs impose substantial socioeconomic strain on families and public systems. Elucidating NDDs’ pathogenesis and advancing therapeutic strategies thus emerge as a critical imperative for safeguarding health in aging societies.

Despite incremental advancements in NDDs’ research, the etiological underpinnings and pathogenic mechanisms remain incompletely elucidated. Current therapeutic interventions are limited in efficacy, lacking disease-modifying agents capable of halting or substantially slowing progression. Conventional pharmaceuticals, while widely used, are frequently costly and associated with adverse effects that compromise long-term adherence [4]. The multifactorial pathogenesis of NDDs—involving intertwined genetic, molecular, and environmental pathways—demands innovative therapeutic strategies. Single-target drug design has shown limited success, whereas multi-target approaches, which address the complexity of NDD pathology, hold significant promise [5].

Traditional Chinese medicine (TCM) offers a compelling paradigm in this context, leveraging its inherent polypharmacological properties to modulate diverse biological targets and pathways [6,7]. *Thesium chinense Turcz*., a medicinal herb utilized in TCM, exemplifies this potential. Its extract contains various components such as flavonoids, polysaccharides, alkaloids, organic acids, and steroids. Recent studies have shown that these components play roles in anti-inflammatory, antioxidant, immune regulation, and neuroprotection [8,9]. This review synthesizes current evidence on the *Thesium chinense Turcz*.’s pharmacologically active compounds and elucidates their therapeutic potential in mitigating progression of NDDs. By delineating its capacity to modulate neuroinflammatory cascades, oxidative stress, apoptosis, and brain–gut axis dynamics [8,9,10,11], this work underscores the viability of *Thesium chinense Turcz*. as a foundation for developing standardized, natural therapeutic agents. Such insights provide a critical translational framework for advancing botanical derivatives into clinically validated NDD treatments, bridging traditional medicinal knowledge with modern drug discovery paradigms. This article reviews the therapeutic potential of *Thesium chinense Turcz*. in alleviating NDDs, providing a theoretical basis for the treatment of NDDs using traditional Chinese medicine.

## 2. The Botanical Characteristics and Chemical Components of *Thesium chinense Turcz.*

*Thesium chinense Turcz.*, a member of the Santalaceae family, is a perennial herb or subshrub characterized by parasitic or hemiparasitic growth habits [12]. Traditionally referred to as “Hundred-milk Grass” or “Fine Beard Grass”, this species thrives in shaded, moist environments such as stream banks, meadows, and agricultural fields, though it also adapts to arid ecosystems including sandy deserts, rocky slopes, and oak forest margins. With a broad geographic distribution spanning tropical and temperate zones, the genus Thesium comprises approximately 400 species across 30 genera globally. In China, 8 genera, 35 species, and 6 varieties have been documented, with *Thesium chinense Turcz.* exhibiting extensive proliferation across 30 provinces—predominantly within temperate and subtropical monsoon climatic regions (e.g., Southwest, Northeast, East, North, and Central China) [13]. This widespread availability ensures cost-effective sourcing of raw materials for pharmaceutical applications.

Renowned in ethnopharmacology as the “phytotherapeutic antibiotic”, *Thesium chinense Turcz.* has been empirically validated for its polypharmacological potential, demonstrating antioxidant, anti-inflammatory, antiviral, anticoagulant, and immunomodulatory properties [14,15]. Such multifunctionality positions it as a versatile candidate for integrative medicinal development, bridging traditional herbal wisdom with contemporary therapeutic innovation.

To date, 34 distinct chemical constituents have been isolated and characterized from *Thesium chinense Turcz.*, encompassing flavonoids, polysaccharides, alkaloids, organic acids, steroids, volatile oils, and related phytochemicals [16]. Both crude extracts and purified fractions of this plant, alongside its commercially available derivatives, exhibit broad-spectrum pharmacological properties—including anti-inflammatory, antimicrobial, analgesic, and hepatoprotective activities—demonstrated across in vitro and in vivo models. These findings underscore its potential as a therapeutic candidate for inflammation-associated pathologies [17]. Notably, flavonoids derived from *Thesium chinense Turcz.* demonstrate dual functionality: beyond their canonical antioxidant capacity, they modulate neurotransmitter dynamics, thereby enhancing neuronal plasticity and synaptic efficacy [18]. Similarly, polysaccharides from this species exhibit pleiotropic bioactivities, such as reactive oxygen species (ROS) scavenging, anti-inflammatory signaling inhibition, and immune homeostasis regulation [19,20,21].

In recent years, *Thesium chinense Turcz.* has emerged as a compelling subject in neuroprotective research, garnering significant attention for its therapeutic versatility. As a naturally derived broad-spectrum antimicrobial agent in TCM, it is clinically employed in diverse formulations—including Baili-Husao granules, tablets, and capsules—for managing multiple pathologies. These preparations leverage well-established pharmaceutical protocols, demonstrating marked clinical efficacy and underscoring the species’ substantial commercial viability [22,23].

## 3. Definition and Development of NDDs

NDDs encompass a spectrum of chronic disorders driven by progressive neuronal degeneration or demyelination [24]. Major NDDs include AD, PD, HD, and ALS. Recent classifications have expanded to incorporate frontotemporal dementia (FTD) and dementia with Lewy bodies (DLB), both pathologically linked to aberrant protein aggregation—such as Transactive Response DNA-Binding Protein 43 kDa (TDP-43) inclusions and α-synuclein deposition [25,26]. These conditions are characterized by irreversible neurological decline, marked by chronic neuroinflammation, motor dysfunction, cognitive deterioration, and behavioral deficits [27,28]. Aging remains the predominant risk factor for NDD onset, and with global demographic shifts toward older populations, the prevalence of these disorders is escalating alarmingly. This trajectory underscores the critical imperative to elucidate pathogenic mechanisms and translate these insights into effective therapeutic or preventive strategies capable of mitigating the societal burden of NDDs.

NDDs typically exhibit an insidious progression, with early-stage manifestations often subclinical or diagnostically elusive. Their etiopathogenesis involves multifactorial determinants, intertwining genetic predispositions with environmental and lifestyle influences. Chronic stressors, sedentary habits, poor nutrition, and airborne pollutants have all been implicated as modifiable risk amplifiers in NDD development [29,30,31]. Current clinical management focuses on symptomatic relief and disease-modifying strategies, including pharmacotherapy (e.g., donepezil, rivastigmine, and memantine for AD), rehabilitation protocols, and surgical interventions. Emerging approaches, such as dietary modulation, show transient efficacy in alleviating cognitive deficits and improving patient quality of life [32]. However, conventional therapeutics—primarily Western pharmaceuticals—face critical limitations: exorbitant costs, adverse side-effect profiles, and an inability to halt pathological progression, rendering them unsuitable for sustained use [33,34]. While these interventions provide palliative benefits, their incapacity to address root pathogenic mechanisms underscores the pressing need for innovative, cost-effective therapies capable of achieving primary prevention or disease modification.

Recent years have witnessed significant advancements in NDD research, with investigators actively pioneering innovative therapeutic modalities across diverse domains. Cutting-edge approaches under exploration include next-generation drug development, stem cell-based regenerative therapies, precision gene editing, ribosomopathy-related protein quality control mechanisms, hyperbaric oxygen interventions, and gut microbiota modulation [35,36,37]. Notably, breakthroughs in computational biology have yielded specialized RNA-editing database systems tailored for NDDs, offering robust bioinformatic infrastructure to accelerate mechanistic and translational studies [38].

Accumulating evidence underscores the therapeutic potential of TCM in mitigating NDD progression through multi-target modulation of pathogenic pathways. Mechanistic studies reveal that TCM-derived compounds regulate neuroinflammatory cascades, pyroptosis, autophagy, oxidative stress, and brain–gut axis dynamics, thereby ameliorating cerebral tissue damage, cognitive deficits, and neuronal loss [39,40,41]. Key bioactive constituents—including flavonoids, polysaccharides, coumarins, salidroside, ginsenosides, ferulic acid, and ligustrizine—exert neuroprotective effects via pleiotropic mechanisms: suppressing neuroinflammation, neutralizing reactive oxygen species, inhibiting apoptosis, restoring gut microbiota homeostasis, modulating neurotransmitter balance, enhancing mitochondrial bioenergetics, improving blood–brain barrier integrity, and regulating insulin signaling pathways [42,43,44]. Notably, TCM formulations demonstrate synergistic polypharmacological actions with fewer adverse effects compared to conventional single-target therapies, positioning them as cost-effective candidates for NDD intervention.

## 4. The Active Ingredients in *Thesium chinense Turcz.* May Affect NDDs

### 4.1. Flavonoids

Emerging research highlights the multifaceted therapeutic potential of flavonoids, demonstrating potent anti-inflammatory, analgesic, antioxidant, and anti-apoptotic properties. *Thesium chinense Turcz.*, a medicinal herb, is particularly enriched with bioactive flavonoids such as kaempferol (Figure 1A), rutin (Figure 1B), and flavone glycosides (Figure 1C), which collectively contribute to its neuroprotective efficacy [15]. These compounds exhibit robust antioxidant activity by scavenging free radicals and inhibiting their generation, thereby mitigating oxidative damage implicated in NDDs [45]. Critical to their anti-inflammatory action is the structural configuration of flavonoids: a planar unsaturated C2–C3 bond and strategically positioned hydroxyl groups, which enhance their capacity to modulate inflammatory cascades [46]. Preclinical and clinical studies further reveal that flavonoid-rich botanicals improve cognitive performance, promote neuronal repair, and stimulate neurogenesis, underscoring their role in restoring neural plasticity [47]. Specifically, flavonoids ameliorate NDD progression through dual mechanisms—suppressing chronic neuroinflammation and directly shielding neurons from oxidative and apoptotic insults. Concurrently, they enhance cerebrovascular function and synaptic plasticity, thereby improving memory and learning deficits.

### 4.2. Polysaccharides

Phytochemical analyses reveal that polysaccharides derived from *Thesium chinense Turcz.* are heteropolymers composed predominantly of mannose, ribose, glucuronic acid, galacturonic acid, glucose, galactose, xylose, arabinose, and fucose, with D-mannose (Figure 1D) and D-glucuronic acid (Figure 1E) serving as the primary monosaccharide constituents [48]. As bioactive agents in TCM, polysaccharides demonstrate preventive, therapeutic, and neurorestorative effects through multi-target mechanisms. These include: (1) suppression of apoptotic pathways, (2) enhancement of neuronal viability via oxidative stress mitigation, (3) attenuation of excitotoxic neuronal damage, and (4) modulation of gut microbiota–brain axis crosstalk [49]. In vitro assays demonstrate that both crude polysaccharides and their three purified fractions possess significant free radical scavenging activity, effectively neutralizing hydroxyl radicals (OH), superoxide anions (O_2_^−^), and 2,2-diphenyl-1-picrylhydrazyl (DPPH) radicals, thereby underscoring their potential to counteract oxidative neurodegeneration [50].

### 4.3. Alkaloids

Alkaloids, a class of nitrogen-containing heterocyclic compounds ubiquitous in TCM, exhibit remarkable structural diversity and broad-spectrum bioactivity. Phytochemical investigations of *Thesium chinense Turc.* have identified N-methylcytisine (Figure 1F), lupanine (Figure 1G), and sophocarpine (Figure 1H) as its principal alkaloidal constituents [51]. These compounds demonstrate therapeutic efficacy across multiple physiological domains, including antibacterial, anti-inflammatory, antiviral, antitumor, antioxidant, and immunomodulatory activities [52,53]. Notably, acetylcholinesterase (AChE) inhibitors—a subclass of alkaloids—ameliorate cholinergic deficits by blocking acetylcholine hydrolysis, thereby elevating synaptic acetylcholine levels. This mechanism underpins their clinical utility in AD management [54]. Mounting evidence suggests that TCM-derived alkaloids enhance central nervous system (CNS) function through multi-target interactions, positioning them as promising candidates for NDD intervention [55]. For instance, huperzine A, a lycopodium alkaloid isolated from Huperzia serrata (commonly known as “snake’s foot stone”), has emerged as a second-generation AChE inhibitor with validated efficacy in treating AD and myasthenia gravis, now recognized in international pharmacopeias [56].

### 4.4. Organic Acids

Phytochemical studies of *Thesium chinense Turcz.* have identified organic acids such as succinic acid (Figure 1I), 4-hydroxybenzoic acid (Figure 1J), 3-p-coumaroylquinic acid, and methyl chlorogenic acid as key bioactive constituents [9]. Structurally defined by their carboxyl (-COOH) functional groups, organic acids are ubiquitous in sour-tasting TCMs and exhibit broad-spectrum pharmacological properties. These include anti-inflammatory responses, inhibition of platelet aggregation, antithrombotic effects, antioxidant activity, and pro-apoptotic actions against tumor cells [57]. Of particular relevance to NDDs are the anti-inflammatory and antioxidant mechanisms of organic acids. By attenuating chronic neuroinflammation and neutralizing reactive oxygen species (ROS), these compounds demonstrate therapeutic potential for mitigating neuronal oxidative damage and inflammation-driven neurodegeneration, positioning them as promising candidates for NDD-focused drug development.

### 4.5. Steroid

*Thesium chinense Turcz.*, a medicinal herb renowned for its pharmacological versatility, contains two principal steroid compounds: periplogenin (Figure 1K) and periplocin (Figure 1L) [14]. Steroids, a class of naturally occurring lipids ubiquitous in biological systems, play multifunctional roles as structural components of cell membranes, physiological regulators, and signaling mediators [58].

Periplogenin, a cardiac glycoside, exhibits anti-inflammatory, antioxidant, and immunomodulatory properties, positioning it as a potential therapeutic agent for immune dysregulation and oxidative stress-related pathologies [59]. Beyond its immunoenhancing capacity to bolster host defenses against pathogens, periplogenin demonstrates marked antineoplastic activity. In vitro studies reveal its ability to suppress colon cancer cell proliferation and induce apoptosis through caspase-dependent pathways, highlighting its translational potential in oncology [60].

Periplocin, another bioactive cardiac glycoside isolated from *Thesium chinense Turcz.*, exerts potent antitumor effects by activating pro-apoptotic signaling cascades and inhibiting tumorigenic progression across multiple cancer types [61]. Its broad-spectrum efficacy against malignancies stems from selective modulation of apoptosis-related pathways, such as p53 activation and Bcl-2 suppression, thereby curtailing uncontrolled cell survival. This mechanistic versatility underscores periplocin’s promise as a scaffold for developing next-generation chemotherapeutics.

### 4.6. Volatile Oil

Volatile oils are lipid-soluble, vaporizable compounds biosynthesized in specialized plant tissues. As pivotal bioactive constituents of natural pharmacopeias, these oils exhibit a ubiquitous presence in botanical taxa, with Chinese flora alone encompassing 136 genera across 56 families confirmed to synthesize such metabolites [62]. Their pleiotropic pharmacological activities span antitussive, antibacterial, anti-inflammatory, analgesic, spasmolytic, anthelmintic, and antitumor effects, alongside cardioprotective, enzyme-modulatory, and antiallergic properties. Particularly relevant to NDD therapeutics are their dual capacities to neutralize reactive oxygen species (ROS) and regulate inflammatory cascades. By attenuating oxidative stress and neuroinflammation—hallmark pathogenic drivers of NDD progression—volatile oils represent a promising reservoir for developing multi-target interventions against neurodegeneration.

## 5. Potential Mechanisms of *Thesium chinense Turcz.* in Modulating NDD Pathogenesis

### 5.1. Anti-Inflammatory

A growing body of research indicates that neuroinflammation plays a pivotal role in the pathogenesis of NDDs [63]. Microglial inflammatory activation is a hallmark of several CNS disorders [64]. Current evidence suggests that the activation of microglia in the CNS and aberrant peripheral innate immune responses precede the onset of clinical symptoms in NDDs. This observation implies that neuroinflammation may serve as a critical mechanistic driver and a key etiological factor in NDDs. Notably, flavonoid compounds have been shown to suppress the production of pro-inflammatory cytokines in activated glial cells. For instance, the flavonoid compound rutin can block activation of the nuclear factor kappa B (NF-κB) signaling pathway and reduce the release of inflammatory cytokines, such as IL-1β, TNF-α, and IL-6 [65].

Moreover, multiple studies have delineated the role of NF-κB signaling in NDDs. Toll-like receptor (TLR) activation induces NF-κB initiation and subsequent release of inflammatory factors, leading to kinase activation, tau hyperphosphorylation, and ultimately protein aggregation, cellular dysfunction, and neurodegeneration [66]. Recently, research by Debashis Dutta and colleagues demonstrated that precise targeting of the TLR2/NF-κB signaling pathway effectively suppresses abnormal α-synuclein propagation both in vitro and in vivo, thereby offering a potential therapeutic strategy for Parkinson’s disease and related neurodegenerative disorders [67]. Periplocin exemplifies anti-inflammatory properties by inhibiting the NF-κB signaling pathway to reduce cell viability and the expression of cytokines (IL-1β and IL-6) [68].

Studies have revealed that Kaempferol, a flavonoid compound, can inhibit the PI3K/AKT signaling pathway [69], suppressing cellular inflammatory responses and extracellular matrix damage. D-mannose, a polysaccharide component, significantly reduces pro-inflammatory macrophages while increasing anti-inflammatory macrophages, thereby attenuating pro-inflammatory responses and enhancing anti-inflammatory effects [70]. In addition, glucuronic acid has anti-inflammatory effects [71], thereby exhibiting therapeutic potential for NDDs [72]. Furthermore, N-Methylcytisine [73,74], Sophocarpine [75], and Succinic Acid [76] have anti-inflammatory biological activity. Periplogenin exerts anti-inflammatory and analgesic effects on diseases by inhibiting cell growth and migration [77]. 4-Hydroxybenzoic acid itself has antioxidant activity and can also inhibit the initiation and activation of Nlrp3 inflammasome [78]. These components may act synergistically with flavonoids to suppress the production of pro-inflammatory cytokines, collectively enhancing their anti-inflammatory efficacy. Building upon this foundation, *Thesium chinense Turcz.* may help mitigate neuroinflammation, thereby emerging as a potential key factor in ameliorating NDDs.

### 5.2. Modulating the Gut Microbiota

Gut–Brain Axis Dysregulation is the cause of the occurrence and development of NDDs. Emerging evidence highlights a robust bidirectional interplay between gut microbiota dysbiosis and neurodegenerative disorders, including PD [79] and neuropsychiatric conditions. Perturbations in the intestinal microenvironment—marked by microbial imbalance, metabolic dysfunction, and compromised barrier integrity—have been implicated in the onset and progression of CNS pathologies. The gut microbiota communicates with the CNS via immune modulation, endocrine signaling, neurotransmitter synthesis (e.g., serotonin, GABA), metabolite production (e.g., short-chain fatty acids), and vagal nerve pathways, collectively contributing to NDD pathogenesis such as AD, PD, and epilepsy [80].

Plant-derived polysaccharides, such as those in *Thesium chinense Turcz.*, exhibit prebiotic properties that reshape gut microbial ecology. These compounds enhance beneficial taxa (e.g., Bifidobacterium, Lactobacillus), suppress pathogenic overgrowth, and fortify intestinal mucosal integrity through mucin synthesis and tight junction stabilization [81]. By restoring microbial homeostasis, *Thesium chinense Turcz.* polysaccharides may attenuate neuroinflammation and oxidative stress via the gut–brain axis, thereby offering a novel therapeutic avenue for NDD intervention.

### 5.3. Against Oxidative Stress

Oxidative stress is implicated in the pathogenesis of various diseases and has been mechanistically linked to the onset and/or progression of NDDs [82]. A hallmark pathological feature of NDDs is sustained oxidative stress (OS), which arises from dysregulated production of reactive oxygen species (ROS) [83]. Notably, mitochondrial dysfunction in neurodegenerative conditions is strongly associated with persistent ROS overgeneration. Therapeutic administration of antioxidants represents a viable strategy for attenuating NDD pathogenesis [84,85].

Investigating antioxidant compounds to rectify the fundamental oxidative/antioxidant imbalance in NDD patients has emerged as a prominent research focus. As previously discussed, both the crude polysaccharides and flavonoids isolated from *Thesium chinense Turcz.* exhibit potent antioxidant properties. The flavonoid glycosides in flavonoids were confirmed to have certain antioxidant activity by DPPH free radical scavenging test [86,87]. Furthermore, steroidal compounds, alkaloids, organic acids, and volatile oils derived from this herb have also been confirmed to possess marked anti-inflammatory and antioxidant activities. For example, Sophocarpine, Succinic Acid, and 4-Hydroxybenzoic acid all have certain antioxidant activities [88]. Through the synergistic actions of these bioactive constituents, *Thesium chinense Turcz.* may ameliorate neurodegenerative pathology by modulating antioxidant stress response pathways, thereby offering a potential therapeutic mechanism for NDD intervention.

### 5.4. Modulation of the Cholinergic Nervous System

Clinically, cognitive decline in AD is positively correlated with reduced levels of the neurotransmitter ACh in the brain, underscoring the therapeutic rationale for acetylcholinesterase (AChE) inhibition to preserve synaptic ACh levels and mitigate symptomatology [89,90]. Emerging studies highlight TCM-derived alkaloids and volatile oils as potent AChE inhibitors [91,92]. *Thesium chinense Turcz.*, for instance, contains bioactive agents with dual neuroprotective mechanisms: (1) Lupanine: Activates nicotinic acetylcholine receptors (nAChRs), attenuating Aβ-induced neurotoxicity (57 ± 2% reduction) and enhancing synaptic plasticity via increased spontaneous calcium transient frequency (60 ± 4% elevation), thereby restoring neural network dynamics in AD models [93]. (2) 4-Hydroxybenzoic Acid: Normalizes coenzyme Q (CoQ) levels, rescuing mitochondrial bioenergetics and ameliorating neurodevelopmental deficits implicated in AD progression [94]. By modulating cholinergic neurotransmission and mitochondrial function, *Thesium chinense Turcz.* presents a multi-target strategy to counteract AD pathogenesis. Its capacity to synergistically enhance ACh signaling while mitigating Aβ toxicity and metabolic dysfunction positions it as a promising candidate for neurodegenerative disease (NDDs) therapeutics.

### 5.5. Modulation of Cerebrovascular Function

Adequate cerebral blood flow (CBF) constitutes a fundamental physiological requirement for maintaining normal brain function. Substantial clinical evidence demonstrates that patients with NDDs consistently exhibit diminished CBF compared to age-matched controls. Compelling epidemiological and clinicopathological data have established significant pathophysiological links between cerebrovascular disease (CVD) and AD, suggesting these conditions may exert cumulative or synergistic effects on accelerating cognitive decline [95]. Of particular therapeutic interest, plant-derived flavonoids exhibit unique neuroprotective properties in experimental studies. These compounds demonstrate potential preventive and therapeutic value for stroke-related brain injury and vascular cognitive impairment, primarily through their ability to enhance cerebral microcirculatory function [96]. Rutin exhibits high affinity for the ACE2 receptor on cell membranes, activates ACE2/Ang1-7 signaling, reduces inflammation, and promotes angiogenesis in the penumbra as well as physiological neovascularization [97]. In addition, the alkaloid Sophocarpine, which is found in *Thesium chinense Turcz.*, has been shown to improve brain damage, such as stroke and cell apoptosis reduction, accompanied by an improvement in neurological scores [98].

### 5.6. Neuroprotective Function

The health and function of neurons are crucial for the normal physiological activities of organisms, and cognitive functions such as learning and memory rely on the efficient operation of the nervous system. Studies have found that flavonoids such as Rutin have significant neuroprotective functions, providing potential strategies for improving neurodegenerative disease states [99,100]. Lupanine in alkaloids has an enhancing effect on neural network activity and synaptic activity, which helps to improve AD and has a neuroprotective effect on it [93]. Meanwhile, Sophocarpine alleviates cognitive impairment and promotes neurogenesis in mouse models of Alzheimer’s disease [101]. Sophocarpine and succinic acid also have anti-apoptotic effects and play a certain degree of neuroprotective function [88,102]. These components achieve neuroprotective effects through multiple pathways, such as inhibiting neuronal apoptosis, maintaining the survival of neurons; regulating neuroinflammatory-related signal pathways to reduce inflammation damage to neurons; enhancing the activity of antioxidant enzymes, eliminating free radicals, and reducing oxidative stress damage to neurons. In addition, they may also regulate synaptic plasticity, promote neuronal connections and communication, help maintain the stability of damaged neural networks, and thus protect and improve neuronal damage in neurodegenerative diseases.

## 6. Discussion

*Thesium chinense Turcz.*, a natural botanical, contains diverse bioactive constituents (e.g., flavonoids, polysaccharides, alkaloids, organic acids, steroids). Current research indicates that these components exert therapeutic effects in anti-inflammation, antioxidation, immunomodulation, and neuroprotection [103]. The neuroprotective actions of its flavonoid compounds involve multiple cerebral mechanisms, including shielding neurons from neurotoxin-induced damage; suppressing neuroinflammation; and enhancing memory, learning, and cognitive function [104]. Polysaccharides from this herb demonstrate well-documented antioxidant and anti-inflammatory activities [19]. Collectively, these antioxidative, anti-inflammatory, and neuroprotective properties highlight its significant potential for preventing and treating NDDs such as AD and PD. The herb may preserve neuronal integrity by reducing intracellular oxidative damage; attenuate neuroinflammatory responses by inhibiting pro-inflammatory cytokines (e.g., TNF-α, IL-6); and promote neuronal repair and regeneration via upregulation of brain-derived neurotrophic factor (BDNF) expression [105,106]. Alkaloids enhance immune regulation to improve CNS function [107]. Organic acids alleviate chronic neuroinflammation through anti-inflammatory and antioxidant effects, thereby ameliorating NDDs [57,108]. Steroids aim to improve and treat NDDs by counteracting immune dysregulation, reducing oxidative stress, and related mechanisms [61,109]. In conclusion, *Thesium chinense Turcz.* can exert beneficial effects on neurodegenerative diseases through multiple pathways and targets (Table 1).

However, current research remains at an early stage. Future studies should employ molecular and cellular biology techniques to elucidate *Thesium chinense Turcz.*’s neuroprotective mechanisms; rigorously evaluate its efficacy and safety in NDDs patients to establish a robust foundation for clinical application; and systematically explore potential synergistic effects when combined with other therapeutics to improve overall patient prognosis.

## 7. Conclusions

*Thesium chinense Turcz.*, a natural botanical rich in bioactive compounds, demonstrates significant potential in the prevention and treatment of NDDs. Its multifaceted mechanisms of action include anti-inflammatory properties, alleviation of oxidative stress, modulation of gut microbiota homeostasis, inhibition of cholinesterase activity, enhancement of cerebrovascular function, and exertion of neuroprotective effects (Figure 2).

These aforementioned synergistic pathways underscore *Thesium chinense Turcz.*’s therapeutic versatility, offering a robust theoretical foundation for advancing natural product research in NDD intervention. This evidence positions it as a promising candidate for developing novel phytopharmaceuticals targeting neurodegenerative pathologies.

## Figures and Tables

**Figure 1 ijms-26-07079-f001:**
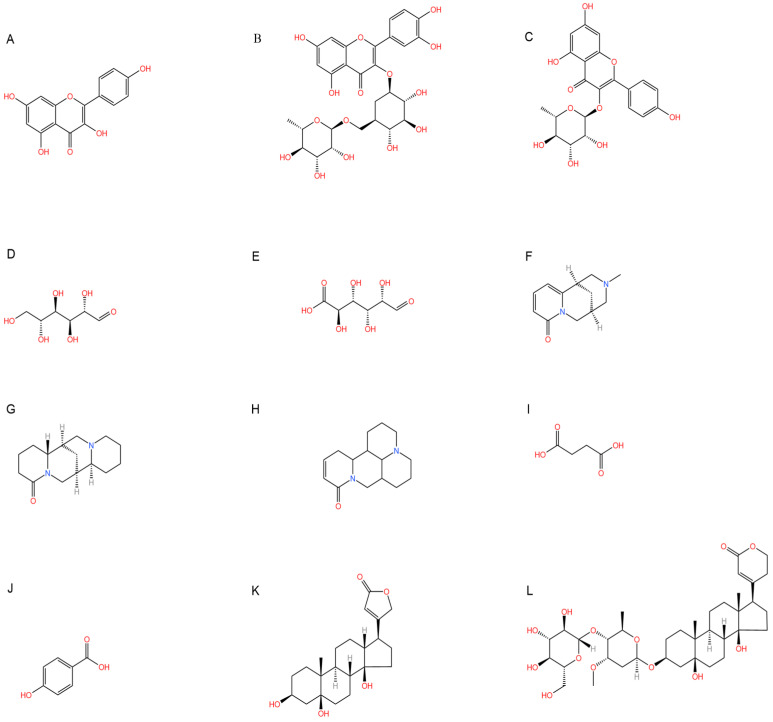
Chemical structure of the main components in *Thesium chinense Turcz.* (**A**) Kaempferol; (**B**) Rutin; (**C**) flavone glycoside; (**D**) D(+)-Mannose; (**E**) Sodium D-Glucuronate Monohydrate; (**F**) N-MMethylcytisine; (**G**) Lupanine; (**H**) Sophocarpine; (**I**) Succinic Acid; (**J**) 4-Hydroxybenzoic acid; (**K**) Periplogenin; (**L**) Periplocin.

**Figure 2 ijms-26-07079-f002:**
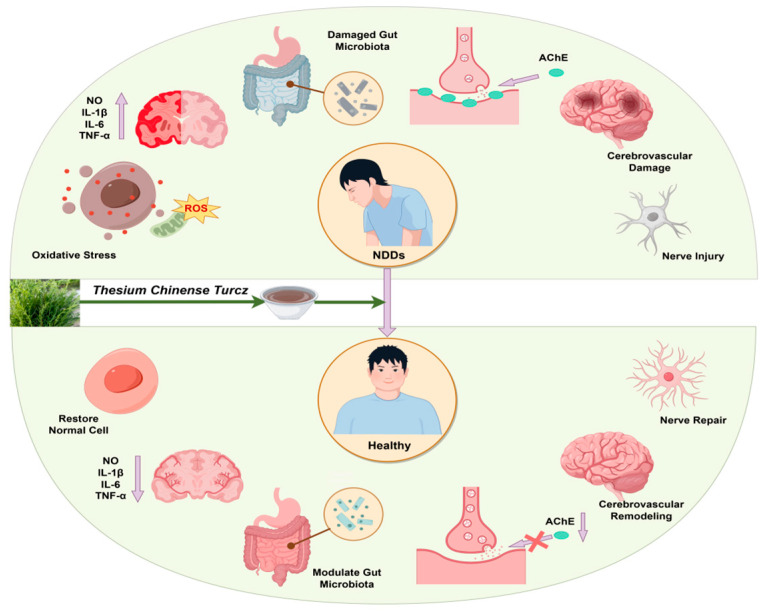
Schematic diagram of the effect of *Thesium chinense Turcz.* on NDDs.

**Table 1 ijms-26-07079-t001:** Bioactive components in *Thesium chinense Turcz.* that may affect NDDs.

Categories	Main Ingredients	Chemical Structure (Molecular Formula)	Role	References
Flavonoids	Kaempferol	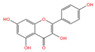	anti-inflammatory and analgesic effects (the hydroxyl position); antioxidant effects and anti-apoptotic effects	[8,14,45,46,47]
Rutin	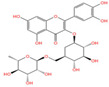
flavone glycoside	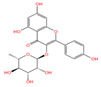
Polysaccharides	D(+)-Mannose	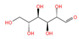	neuroprotective effects (apoptosis, enhancing the vitality of nerve cells and resisting oxidative stress injury, and inhibiting excitotoxicity to alleviate further damage of neurons) and andantioxidant effects influence the regulation of the intestinal flora	[48,49,50]
Sodium D-Glucuronate Monohydrate	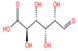
Alkaloids	N-Methylcytisine	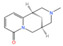	antibacterial, anti-inflammatory, antiviral, anti-tumor, antioxidant, and immune regulation improve the central nervous system	[51,52,54,55,56]
Lupanine	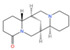
Sophocarpine	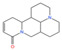
Organic acids	Succinic Acid	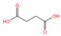	anti-inflammatory responses, inhibition of platelet aggregation, anti-thrombosis, antioxidation, and induction of tumor cell apoptosis, etc.	[34,57]
4-Hydroxybenzoicacid	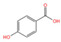
The steroid compounds	Periplogenin	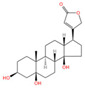	anti-inflammatory, antioxidant, and immunomodulatory effects effectively enhance the immune function of the body and help resist the invasion of external pathogens; anti-colon cancer effect	[59,60]
periplocin	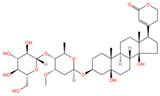	promote apoptosis of tumor cells and inhibit tumor growth (anti-tumor mechanism may be related to the activation of apoptosis-related signaling pathways)	[61]

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
