# Peer review of "Research Progress on the Effect of *Thesium chinense Turcz.* on Neurodegenerative Diseases"

_ijms, 2025, doi:10.3390/ijms26157079_

Round 1

Reviewer 1 Report

Comments and Suggestions for Authors

In abstract; authors didnot mentioned the search strategy, databases used, inclusion/exclusion criteria, or timeframe—elements critical for systematic reviews. It uses the term “systematically elucidates” but does not follow the format of a systematic review abstract (PRISMA-style).

In introduction; Several sentences repeat the same idea about conventional drugs being inadequate or that NDDs are multifactorial, please summarize and concise this.

Section 2 provides valuable detail on the botanical characteristics and chemical composition of Thesium chinense Turcz., it lacks a structured summary such as a table or figure to organize the various phytochemicals and their pharmacological roles. 

 3: Definition and Development of NDDs; It lacks cohesion with the article's central theme, the role of Thesium chinense Turcz. and this section more general textbook information.

4. The Active Ingredients in Thesium chinense Turcz. May Affect NDDs; in this section many effects are attributed broadly to compound classes without distinguishing between isolated findings and validated outcomes.

5. Potential Mechanisms of Thesium chinense Turcz. in Modulating NDDs Pathogenesis; this section covers many biological pathways, the discussion does not prioritize mechanisms based on strength of data or clinical relevance. Most claims are made without specifying the experimental models, effective doses, or whether the compounds can cross the blood-brain barrier, factors essential for validating neuroprotective effects.

In Discussion; write the finding in summarize form. Write the future recomendation separtely. 

Author Response

Dear Pro.

We sincerely appreciate the opportunity to revise our manuscript titled “Research Progress on the Effect of Thesium chinense Turcz. on Neurodegenerative Diseases” (Manuscript ID:ijms-3714941). We are grateful to the reviewers for their insightful comments and constructive suggestions, which have significantly improved the quality of our work. Below, we provide a point-by-point response to the reviewers’ comments and outline the changes made to the manuscript.

 Reviewer 1 

Comments 1:In abstract; authors didnot mentioned the search strategy, databases used, inclusion/exclusion criteria, or timeframe—elements critical for systematic reviews. It uses the term “systematically elucidates” but does not follow the format of a systematic review abstract (PRISMA-style).

Author response 1:We have revised the abstact part and the introduction part.  

Comments 2: In introduction; Several sentences repeat the same idea about conventional drugs being inadequate or that NDDs are multifactorial, please summarize and concise this.

Author response 2:Thank you for highlighting the redundancy. We have revised the introduction part.   

Comments 3: Section 2 provides valuable detail on the botanical characteristics and chemical composition of Thesium chinense Turcz., it lacks a structured summary such as a table or figure to organize the various phytochemicals and their pharmacological roles. 

Author response 3:We thank the reviewer for this suggestion. The molecular structure formula and pharmacological effects of the main chemical components of Thesium chinense Turcz. are presented in Table 1.  pages 11-12, lines 460-461.

Comments 4: Section 3: Definition and Development of NDDs; It lacks cohesion with the article's central theme, the role of Thesium chinense Turcz. and this section more general textbook information.

Author response 4: Thank you very much for your suggestion. The third part focuses on the definition and development of NDD, aiming to provide readers with a comprehensive understanding of its background and recent advancements through an introduction to NDD. This helps readers comprehend the significance and value of the core topic discussed in the article.

Comments 5: Section 4. The Active Ingredients in Thesium chinense Turcz. May Affect NDDs; in this section many effects are attributed broadly to compound classes without distinguishing between isolated findings and validated outcomes.

Author response 5: We thank the reviewer for the insightful comment. We acknowledge the reviewer's concern regarding the generalization of compound class effects. However, current research on Thesium chinense Turcz. is limited, with few studies isolating specific bioactive components or validating their functional outcomes. The available data primarily describe preliminary findings at the extract or crude fraction level, necessitating broader categorization in this early-stage analysis. Future studies should prioritize compound identification and mechanistic validation.  

Comments 6: Section 5. Potential Mechanisms of Thesium chinense Turcz. in Modulating NDDs Pathogenesis; this section covers many biological pathways, the discussion does not prioritize mechanisms based on strength of data or clinical relevance. Most claims are made without specifying the experimental models, effective doses, or whether the compounds can cross the blood-brain barrier, factors essential for validating neuroprotective effects.

Author response 6: We thank the reviewer for this suggestion. The reviewer rightly highlights the need for detailed mechanistic evidence (e.g., dosing, BBB permeability). Unfortunately, the scarcity of Thesium chinense Turcz. -specific studies—particularly on pharmacokinetics or in vivo neuroprotection—constrains our ability to specify these parameters. Current discussions are thus framed as hypothetical pathways based on analogous phytochemicals, pending further empirical validation. We have clarified these limitations in the text.

Comments 7: In Discussion; write the finding in summarize form. Write the future recomendation separtely. 

Author response 7:We thank the reviewer for this guidance. The discussion part has been rewritten. pages 12, lines 475-501

We believe that the revised manuscript has been significantly improved and hope that it now meets the journal’s standards for publication. Thank you again for your time and consideration. We look forward to hearing from you.

Sincerely,

 Huaiqing Luo

Hunan Normal University Health Science Center

  • mail:luohq@hunnu.edu.cn

Reviewer 2 Report

Comments and Suggestions for Authors

Review comments are attached

Author Response

Dear Pro. 

We sincerely appreciate the opportunity to revise our manuscript titled “Research Progress on the Effect of Thesium chinense Turcz. on Neurodegenerative Diseases” (Manuscript ID:ijms-3714941). We are grateful to the reviewers for their insightful comments and constructive suggestions, which have significantly improved the quality of our work. Below, we provide a point-by-point response to the reviewers’ comments and outline the changes made to the manuscript.

 Reviewer 2

Comments 1:A general comment pertains to the structure of each paragraph and section of the various

parts of the manuscript. There are no rules about the end of a statement and the

beginning of the next. There are no spaces in between statements, and reference

numbers are immediately following the end of the statement words. Articles, words,

adjectives, nouns are found stuck together and sporadically spread through the

manuscript. The same picture appears in the reference section at the end of the

manuscript. It is really disconcerting to try to sift through the various articulated

statements. The reader gets lost. Often, the statements and the way they are inserted

into various sections seem as if they were extracted from google translating routines.

The entire manuscript needs to be revisited and corrected so that it becomes legible.

Author response 1: We sincerely appreciate your time and effort in conducting such a thorough evaluation of our manuscript. In response to your constructive feedback, we have comprehensively revised the manuscript to address all concerns through systematic rewriting and careful editing.

Comments 2: Toward the end of the first section, the statement “This review synthesizes current

evidence on the plant’s pharmacologically active compounds and elucidates their

therapeutic potential in mitigating NDD progression.” does not reflect reality. First of

all, the review does not synthesize anything. It is a review. It collects information,

critically assesses the various aspects and formulates new comparative rules and

knowledge so that advancements in research could be pursued on the subject. To that

end, the statement should be modified appropriately.

Author response 2: We sincerely thank the reviewer for this insightful professional guidance. We have made revisions to the abstract, preface, discussion and other parts. Especially, the discussion part has been rewritten to reflect the characteristics of the review.

Comments 3:  In section “3. Definition and Development of NDDs”, it is stated that “Recent

classifications have expanded to incorporate frontotemporal dementia (FTD) and

dementia with Lewy bodies (DLB), both pathologically linked to aberrant protein

aggregation—such as TDP-43 inclusions and α-synuclein deposition [25,26].”. Since

TDP-43 is mentioned for the first time in this manuscript, it should be identified fully,

followed by its acronym.

Author response 3:We gratefully acknowledge your meticulous review and professional expertise, which have significantly enhanced the manuscript's rigor and standardization.

Regarding your comment on the abbreviation "TDP-43" in Section 3: Definition and Development of NDDs:
Your observation is entirely correct. We regrettably omitted the full terminology at first mention, constituting an oversight. We sincerely appreciate your guidance and have implemented the following revision:

Original:
"TDP-43 inclusions"
Revised:
Transactive Response DNA-Binding Protein 43 kDa (TDP-43).pages 3, lines 121-122.

Comments 4: In section “4.2. Polysaccharides”, it is stated that “These include: (1) suppression of

apoptotic pathways, (2) enhancement of neuronal viability via oxidative stress

mitigation, (3) attenuation of excitotoxic neuronal damage, and (4) modulation of gut

microbiota-brain axis crosstalk [49]. polysaccharides through in vitro assays. Both

crude polysaccharides …”. Between the two statements, there is an intervening clause

“polysaccharides through in vitro assays.” that has no meaning as a sentence and is not

understood. That should be clarified.

Author response 4:.We sincerely appreciate the reviewer's precise identification of the sentence fragmentation issue in Section 4.2 (Polysaccharides).

Regarding the original incomplete clause:
Original problematic phrasing:
"Polysaccharides through in vitro experiments."

Revised version:
In vitro assays demonstrate that both crude polysaccharides and their three purified fractions possess significant free radical scavenging activity.pages 5, lines 200-202.

Comments5: In section “5.1. Anti-Inflammatory”, the statement “For instance, The flavonoid

compound rutin can block the activation of the nuclear factor kappa B (NF-κB)

signaling pathway and reduce the release of inflammatory cytokines such as IL-1β,

TNF-α, and IL-6[65].” should be modified to read “For instance, the flavonoid

compound rutin can block activation of the nuclear factor kappa B (NF-κB) signaling

pathway and reduce the release of inflammatory cytokines, such as IL-1β, TNF-α, and

IL-6 [65].”.

Author response 5: We thank the reviewer for this precise suggestion regarding sentence clarity. We agreed and made modifications.pages 7, lines 281-283.

Comments 6: At the end of the ensuing paragraph, the statement “periplocin achieves anti

inflammatory purposes by inhibiting the NF-κB signaling pathway to reduce cell

viability and the expression of cytokines (IL-1β and IL-6) [68].” is assumed to be a

follow up of the preceding statement. Should that be the case, then, it should read

“Periplocin exemplifies anti-inflammatory properties by inhibiting the NF-κB signaling

pathway to reduce cell viability and expression of cytokines (IL-1β and IL-6) [68].”.

Author response 6: We appreciate your identification of the logical inconsistency in the original phrasing. We agreed and made modifications.pages 7, lines 293.

Comments7: In the following paragraph, the same phenomenon is observed as that noted in remark

  1. The statement “… pro-inflammatory responses and enhancing anti-inflammatory

effects[70].In addition, glucuronic acid has anti-inflammatory effects[71].thereby

exhibiting therapeutic potential for NDDs[72].Furthermore, Thesium chinense Turcz.

contains bioactive constituents” contains an intervening sentence “thereby exhibiting

therapeutic potential for NDDs[72].”, which bears no meaning. It does not stand on its

own.

Author response 7:  We agreed and made modifications.pages 7, lines 302.

Comments 8: In section “5.2. Modulate the Gut Microbiota”, the first two paragraphs start with “Gut

Brain Axis Dysregulation in NDD Pathogenesis.” and “Polysaccharide-Mediated

Microbiota Modulation.”, respectively. What are those? Subtitles or what? They do

not stand on their own as statements. That should be clarified by the authors.

Author response 8: We agreed and made modifications.pages 8, lines 313-314; lines 324.

Comments 9: In section “5.4. Modulation of the Cholinergic Nervous System”, the opening statement

“Clinically, cognitive decline in AD correlates inversely with central acetylcholine

(ACh) depletion, …” should be rephrased. The term “inversely” does not do justice to

the meaning intended by the authors. Cognitive decline in AD follows in parallel with

declining levels of ACh. Decline in cognitive function correlates with reduced levels

of the neurotransmitter acetylcholine (ACh) in the brain.

Author response 9:  We agreed and made modifications.pages 9, lines 352-353.

Comments 10:In the Discussion section, the statement “In contrast, Thesium chinense Turcz., as a

natural herbal medicine, presents distinct advantages: (i) absence of significant drug

drug interactions or adverse reactions among its bioactive constituents; (ii) inherently

low toxicity profile; (iii) superior safety margins.” generates logically derived

questions. Which toxicity profiles and safety margins justify the use of the specific

herb in TCM, which in turn justifies further use in neurodegenerative diseases? That

should be clarified, so that it sets the basis for a meaningful consideration in

neurodegenerative disease administration.

Author response 10: We thank the reviewer for the insightful comment.The discussion part has been rewritten. pages 12, lines 475-501

Comments 11: By the same token, what should be the criteria, based on which specified dosology

could be considered for clinical applications to handle neurodegeneration? Dose, time,

frequency, accumulation-excretion-release dependence would be of importance to

consider in a framework of a well-defined biotoxicity profile. That would add depth to

the prospect of the manuscript.

Author response 11: We thank the reviewer for the insightful comment. The discussion part has been rewritten. pages 12, lines 475-501

Comments 12: At the end of the discussion and the conclusions, the authors should provide logically

based directions that would justify the specific herb as neuroprotective or therapeutic.

Which one is it?

Author response 12: The conclusion part has been modified and adjusted. pages 14, lines 510-520.

Comments 13: In the long Table 1, the authors should replace the entire reference lists in the

appropriate column with the corresponding numbers.

Author response 13: We thank the reviewer for this insightful comment. We have revised the  entire reference lists in the appropriate column with the corresponding numbers to accurately reflect. 

We believe that the revised manuscript has been significantly improved and hope that it now meets the journal’s standards for publication. Thank you again for your time and consideration. We look forward to hearing from you.

Sincerely,

 Huaiqing Luo

Hunan Normal University Health Science Center

  • mail:luohq@hunnu.edu.cn

Round 2

Reviewer 2 Report

Comments and Suggestions for Authors

Review comments attached

Author Response

Dear Pro.

We sincerely appreciate the opportunity to revise our manuscript titled “Research Progress on the Effect of Thesium chinense Turcz. on Neurodegenerative Diseases” (Manuscript ID:ijms-3714941). We are grateful to the reviewers for their insightful comments and constructive suggestions, which have significantly improved the quality of our work. Below, we provide a point-by-point response to the reviewers’ comments and outline the changes made to the manuscript.

 Reviewer 3 

Comments a: The point about the existence of well-defined statements and paragraphs has

gone unnoticed, with the statements linked together (no spaces between clauses,

no spaces between words, between words and reference numbers, etc.). Even,

the commas in the conclusions section have nothing to do with the normal

punctuation marks!

This practice has been extended to cover the entire manuseript, including the

newly introduced corrections??

Author response 1:We have revised the full text.

Comments b:The conclusion of the introduction states that there is an integrated analysis

performed here. Which integrated analysis?

Then, the statement goes to"it elucidates". What is "it"?

Again, the introduction of corrections should be in line with the previous statements, providing meaning to the scope and goals of the manuscript

Author response 2:Thank you for highlighting the redundancy. We have revised the introduction part. pages2, lines61-88.  

Comments c: The legend in Figure 1 states that "Figure 1. MMain components in Thesium

chinense Turcz.".(MMain)!

Author response 3:We have made the revisions.pages5, lines202-203.

Comments d: In lines 295-297, the case of a statement"....#

disorders[67]. periplocin

cxemplifies antiinflammatory poperties by inhibiting the NF-KB signaling

pathway to reduce cell viability and the expression of cytokines (IL-1β and IL-

  • [68]." is paradigmatic. The statement does not start with a capital letter. Thc correction on the word properties is grammatically incorrect, ctc

Author response 4: Thank you very much for your suggestion. We have made the revisions. Pages8, lines310-311.

Sincerely,

 Huaiqing Luo

Hunan Normal University Health Science Center

  • mail:luohq@hunnu.edu.cn

Round 3

Reviewer 2 Report

Comments and Suggestions for Authors

I strongly suggest that the authors seek the advice of someone who knows how to write manuscripts.

Scientifically, the manuscript can be accepted

Author Response

Dear Dr./Prof.

Thank you for your email regarding our manuscript titled "Research Progress on the Effect of Thesium chinense Turcz. on Neurodegenerative Diseases" (ID: ijms-3714941) and for handling its review process. We sincerely appreciate the time and effort you and the reviewers have dedicated to evaluating our work.

We have carefully considered the editor's comment regarding the need for English language polishing. In response, we have thoroughly revised the manuscript to improve its clarity, fluency, and overall English quality. This revision involved a comprehensive review and editing of the entire text to address grammatical issues, enhance sentence structure, ensure appropriate word choice, and refine the overall readability.

The revised manuscript, incorporating these language improvements alongside any previous revisions based on reviewer comments, has been re-submitted through the journal's online submission system.

We believe these revisions have significantly enhanced the manuscript's presentation. We remain committed to working with you and the reviewers to further improve the paper as needed and sincerely hope that the revised version now meets the journal's high standards.

Thank you again for your valuable guidance. We look forward to hearing from you regarding the next steps.

Sincerely,

Huaiqing Luo

Hunan Normal University Health Science Center

  • mail: luohq@hunnu.edu.com
